# Next-Generation Molecular Discovery: From Bottom-Up In Vivo and In Vitro Approaches to In Silico Top-Down Approaches for Therapeutics Neogenesis

**DOI:** 10.3390/life12030363

**Published:** 2022-03-02

**Authors:** Sophie E. Kenny, Fiach Antaw, Warwick J. Locke, Christopher B. Howard, Darren Korbie, Matt Trau

**Affiliations:** 1Centre for Personalised Nanomedicine, Australian Institute for Bioengineering and Nanotechnology (AIBN), The University of Queensland, Corner of College and Cooper Roads (Bldg 75), Brisbane, QLD 4072, Australia; sophie.kenny@uq.edu.au (S.E.K.); f.antaw@uq.edu.au (F.A.); c.howard2@uq.edu.au (C.B.H.); 2Molecular Diagnostic Solutions, Health and Biosecurity, Commonwealth Scientific and Industrial Research Organisation, Building 101, Clunies Ross Street, Canberra, ACT 2601, Australia; warwick.locke@csiro.au; 3School of Chemistry and Molecular Biosciences, The University of Queensland, Brisbane, QLD 4072, Australia

**Keywords:** therapeutics, artificial intelligence, drug development, high-throughput library screening, antibody and peptide discovery, chemical libraries, phage display, protein folding prediction

## Abstract

Protein and drug engineering comprises a major part of the medical and research industries, and yet approaches to discovering and understanding therapeutic molecular interactions in biological systems rely on trial and error. The general approach to molecular discovery involves screening large libraries of compounds, proteins, or antibodies, or in vivo antibody generation, which could be considered “bottom-up” approaches to therapeutic discovery. In these bottom-up approaches, a minimal amount is known about the therapeutics at the start of the process, but through meticulous and exhaustive laboratory work, the molecule is characterised in detail. In contrast, the advent of “big data” and access to extensive online databases and machine learning technologies offers promising new avenues to understanding molecular interactions. Artificial intelligence (AI) now has the potential to predict protein structure at an unprecedented accuracy using only the genetic sequence. This predictive approach to characterising molecular structure—when accompanied by high-quality experimental data for model training—has the capacity to invert the process of molecular discovery and characterisation. The process has potential to be transformed into a top-down approach, where new molecules can be designed directly based on the structure of a target and the desired function, rather than performing screening of large libraries of molecular variants. This paper will provide a brief evaluation of bottom-up approaches to discovering and characterising biological molecules and will discuss recent advances towards developing top-down approaches and the prospects of this.

## 1. Introduction

Understanding and predicting molecular interactions in living systems and designing new molecules to bind to, destroy, disrupt, or promote biological molecular pathways have long been driving forces behind much of molecular biology research and funding. In recent years, the therapeutics industry has boomed. In 2020, the FDA approved 53 drugs, the second highest approval rate ever [1], and the global pharmaceutical market was estimated to be worth at least USD 1.2 trillion in 2019 [2]. Whilst about 75% of the market is composed of small-molecule drugs, with only 20% biologics and 5% peptides, these markets are expanding, with the monoclonal antibody (mAb) market estimated to reach USD 130–200 billion in 2022 [3]. Interest in the field is massive, and rapidly growing.

The novel coronavirus pandemic has highlighted the urgency for efficient discovery and production of molecular therapeutics. For example, the first drug to be approved across major global regulatory authorities for the antiviral treatment of COVID-19 was remdesivir (see Table 1), where it took approximately one year for other antiviral drugs to receive authorisation. In Australia, it took about 6 months for remdesivir to be approved after the first wave of cases arrived. Additionally, it took over 1.5 years in Australia to see the approval of any other COVID-19 antiviral drug other than remdesivir, where no other antiviral treatment options were available for use until August 2021. In the United States, remdesivir was the only non-emergency use antiviral drug approved by the FDA until the authorisation of baricitinib on 19 November 2020 [4]. Even the use of human plasma—collected from individuals with high titres of anti-SARS-CoV-2 antibodies—was approved (23 August 2020) before baricitinib. The earlier authorisation of remdesivir globally for SARS-CoV-2 treatment in comparison to other antiviral drugs was only possible due to long-term testing on other RNA viruses such as Ebola and other coronaviruses in the years leading up to the pandemic [5,6]. Whilst remdesivir has been shown to be an effective anti-viral agent [7] (although this has been strongly debated [8]), the drug requires IV administration. This means it is not readily given to the general population for mild infections that do not require hospitalisation. In addition, as with most antiviral or antibacterial molecules, there is some possibility for viral resistance to arise [9], which means that having alternative treatments is optimal. The current slow process for therapeutic discovery and the inherent risk in having such a limited suite of anti-viral options prompts the question of how faster pipelines for molecular therapeutics development can be achieved.

Advances in molecular discovery have traditionally focussed on improving the efficiency and reliability of in vitro and in vivo discovery and synthesis techniques, approaches which can be considered as “bottom-up” techniques. In this review we specifically define “bottom-up” as methods for therapeutic molecular discovery that use animal antibody generation (in vivo), or the screening vast libraries of potential therapeutic molecules against therapeutic targets (in vitro), where molecules with the best biological response are selected (see Figure 1). These approaches are generally slow and arduous and in essence “find what sticks”, or “find the needle in the haystack”. Moreover, because little is known about the new therapeutic after discovery, there are significant resources spent in molecular characterisation and testing before the therapeutic is ready for clinical trials. There have been many approaches to reduce the time and resources spent in current discovery programs, yet there is not any technique that comprehensively turns this process around through a “top-down” approach, where a specific therapeutic molecule could be built from scratch based directly on the therapeutic target and desired outcome. Ideally, a top-down approach would eliminate the need to perform large library screening campaigns or perform in vivo antibody generation. It would allow for much higher confidence in target-binding specificity through a process of screening “self” molecules and therefore reduce the potential for side-effects in patients. Not only would this reduce risk for patients, but also investors, where financial investment is often one of the bottlenecks for taking a potentially life-saving drug to the market, and therefore deliverable to the patient. In this review we will explore the movement towards the top-down approach for molecular discovery, or rather design, and the potential to fast-track the pipeline for therapeutics from discovery to the clinic.

## 2. Bottom-Up Approaches

Molecular discovery may be broadly grouped into in vivo, in vitro and in silico approaches. Most current approaches to molecular discovery may be considered bottom-up, particularly for in vivo or in vitro approaches, as generally little is known about the conformation of the final therapeutic before screening or antibody generation takes place. Whilst these techniques have not yet succeeded in flipping discovery towards top-down approaches, the field has produced a wealth of innovation and advancements that have enabled significant leaps in the field of therapeutic discovery. The therapeutics discovery pipeline has been outlined in Figure 2, with examples of some of the prominent approaches to each stage of the pipeline. Some of the most recent innovations that contribute to this pipeline will be outlined in the following sections.

### 2.1. In Vivo Techniques

In vivo techniques have typically relied on the animal and human natural adaptive immune response, where antibodies are produced upon immunisation with a desired antigen [10,11,12] (see Figure 3). Many types of animals can be used, although mice, rats, rabbits, guinea pigs, hamsters, chickens, and larger animals such as goats, horses (particularly for antivenoms), camelids and sheep are common. Antibody variation, such as the combinatorial diversification of V(D)J recombination, somatic hypermutation and class switching [10,13,14], enables a high immunoglobulin repertoire, which in humans can reach 10^11^ different molecular combinations [15]. Humanised rodent-produced mAbs were the first mAbs to be approved for therapeutics [10], where production required the immunisation of mice with an antigen of interest, followed by fusion of spleen cells with myeloma cells to form immortalised hybridomas [16]. Hybridomas are then screened and selected based on the desired affinity and specificity of the produced antibody.

There are some limitations to in vivo techniques, but new innovations have sought to overcome these. First, mAbs produced in non-human models traditionally require humanisation in order to minimise undesirable responses from the human immune system [17]. However, the engineering of transgenic mice now allows the production of fully humanised antibodies directly from the animal [10,18,19], as well as techniques for the grafting of the complementarity determining regions of murine antibodies into human antibody frameworks [17]. Recently, the development of camelid nanobody or VHH (variable heavy chain domains of heavy chain antibodies)-producing mice enabled the production of enhanced avidity, neutralising binders for SARS-CoV-2 [20]. It was suggested that these could reduce likelihood of immune evasion by binding to conserved regions of the virus that are usually inaccessible to larger normal antibodies. There have also been human donor approaches to antibody generation. For example, B cells isolated from patients naturally infected with SARS-CoV-2 produced virus-neutralising activity in Syrian hamsters [21]. This enabled estimation of the most common patient antibody response in terms of targeted sites on the virus, particularly ACE2 binding site (competition studies), and isolation and characterisation of neutralising antibodies. However, likewise to antibodies produced in animal models, significant testing in animals is generally still required for human-derived antibodies before progression to clinical trials would be possible.

While traditional in vivo methods benefit from the tremendous natural diversity of the animal immune response, they are not as high throughput as library screening techniques. It can take a significant amount of time (around 6 weeks [22]) for immunised animals to illicit an immune response and produce antigen-targeting B cells before target-binding properties can be screened. In vivo techniques are also generally expensive, requiring experienced laboratory technicians, specialised laboratory facilities, ethics approval and mouse sacrifice. Although, as antigen properties and conformation are affected by the conditions of the solution they reside in, they will be closest to their natural state when presented to the immune system in in vivo. This means that in vivo techniques allow for the generation of antibodies that will bind to antigens in the real biological setting, marking them as an attractive method for therapeutics discovery, despite time and handling issues.

### 2.2. In Vitro Techniques

In vitro approaches were initially aimed towards widening the range of possible therapeutic molecule types and making discovery more easily scalable than in vivo techniques. Techniques are generally geared towards mimicking the natural immune response in vitro and technological advances broadly involve widening the sampling of chemical space through high-throughput wet lab technologies. In addition to now having higher throughput than in vivo approaches, in vitro techniques can allow for screening against toxic compounds that may alternatively have been difficult or cumbersome to produce neutralising antibodies for in vivo approaches. For example, antivenoms can be produced in vitro, rather than using the traditional approach of horse immunisation. One approach used phage-displayed camelid nanobodies to produce neutralising activity against *Bothrops jararacussu* snake venom [23]. High costs of animal maintenance and difficulties in producing antivenom (stability of antivenom, obtaining enough venom from snakes) [24] make in vitro approaches attractive. However, in vitro discovery is still limited by the time taken for wet-lab experimental procedures to be carried out and by molecular library size, and all techniques still fall into the “bottom-up” approach classification.

Most in vitro techniques rely on the production of hugely diverse libraries of chemicals, peptides, antibodies, or antibody fragments, with the aim of isolating unique variants from the combinatory pool that will bind to a desired antigen. This means that larger library diversities are more advantageous as the likelihood of identifying a suitable binder scale proportionally. Libraries can be either labelled (linked to a descriptive tag such as DNA) or unlabelled. Labelled library types include well-established phage display [25,26], bacteria [27,28], yeast [29,30,31], mammalian cell display [32], and cell-free techniques such as ribosome [33], mRNA display [34,35] and DNA-linked chemical libraries [36]. Generally, most biologically generated libraries such as antibody or protein libraries are labelled, whereas unlabelled libraries are composed of individually stored molecules, for example high-throughput chemistry (HTC) [37,38] chemical libraries.

Initial molecular discovery techniques did not involve the use of labelled libraries, and instead relied on the synthesis and individual storage of unique compounds. A limitation in the size of available molecular libraries was, for a significant period, one of the major bottlenecks to new molecule discovery [39]. The advent of HTC techniques has now allowed for the creation of unlabelled chemical libraries of up to 3 million individually stored compounds. To screen these libraries in vitro, each molecule must be examined individually in separate reactions against targets to assess activity. Whilst this process is lengthy and inefficient if performed by hand, the speed of this has been significantly increased by high-throughput screening (HTS). This technology is the use of highly efficient automated robotic platforms to screen each library compound one-by-one against a target in high multiplicity multi-well plates [40,41,42]. HTS is essentially a brute-force solution to the lengthy wet lab procedure, but is still used frequently, particularly for chemical library screening.

Developed by George Smith in 1985, phage display has been one of the driving technologies for the discovery of new antibodies and peptides through labelled molecular library screening [26]. It is one of the original efficient methods for linking a molecular phenotype with its descriptive genotype, which allows for the identification and characterisations of a molecule isolated from a library of molecules during library screening. Phage display is the cloning of a desired variable DNA sequence, either of a peptide [25] or antibody fragment [26,43,44], into the positive-sense ssDNA genome of filamentous bacteriophage, where it is expressed as a protein fusion to one of the M13 bacteriophage coat proteins (generally pIII). These genetically linked molecular libraries can be divided into naïve, immune, semi-synthetic and fully synthetic libraries, depending on the origin of the displayed molecules. Non-immunised and immunised animals provide the source for naïve and immune libraries for antibody or antibody fragment display, respectively, whereas semi-synthetic libraries are sequences modified from animal or human donors. Fully synthetic libraries are those that display completely synthetically constructed complementary determining regions (CDRs) on known antibody frameworks, or synthetically constructed peptides, allowing for the generation of libraries with huge, randomised diversity [45,46]. These genotype-linked molecules can be screened through affinity enrichment, in a process called biopanning (see Figure 4), where molecules that bind to a target of interest are isolated.

There is a range of different labelled approaches (see Table 2) that are used for different purposes. DNA-encoded chemical libraries [36] have enabled expansion of the screening repertoire for up to 10^8^ molecules, where compounds are chemically conjugated to DNA sequences that indicate their chemistry. Cell-free display techniques such as mRNA display allow for the production of libraries, with some of the largest degrees of genetic variation—reportedly above 10^12^ variants—because they can employ PCR-based approaches for generation of library variation [47]. Some techniques such as the protocol designed by Jones et al. (2016) have also performed phage display on cell-presented antigens (such as membrane-bound receptors) [48]. In this system, panning occurs on cells in solution, where the target is presented in its native state, including post-translational modifications.

One notable drawback of cell-free in vitro display methods is that the size and complexity of the molecules that can be expressed are usually limited. For example, phage display is limited by attenuation of phage infectivity and inefficient incorporation of large proteins into the major pIII phage coat protein [59]. Normally the pIII coat protein mediates phage infection of bacterial cells by attachment to the F-pilus [60], and this means that there is limit to the size of the molecule that can cloned into the pIII protein (about 50 residues) before phage infectivity is affected [59]. It also means that it is possible for library diversity to be affected through biopanning by the variable ability of each phage clone to re-infect the bacterial host. Other cell-free techniques such as mRNA display may eliminate this cell infection fitness bias. However, mRNA display is typically only useful for small peptide screening, as there is a reduced display efficiency for proteins larger than about 110 residues [61].

In contrast, cellular techniques such as bacteria, yeast, and mammalian surface display have the advantage of allowing for the display of larger molecules and do not require the post-panning bacterial infection step that may cause infection efficiency bias in enriched phage libraries. However, due to the necessity for cell transformation, cell display library sizes are smaller, reaching about a maximum of about 10^9^ variants [62]. Both cellular and acellular in vitro approaches are generally higher throughput than in vivo methods, because they do not require the lengthy process of mouse sacrifice and spleen cell recovery for myeloid cell fusion. However, because in vitro techniques screen against targets outside the conditions present in normal immune response, antibodies generated against desired antigens may have altered pharmacokinetics and have more likelihood of causing unintended responses.

It is possible that future peptide library panning campaigns will not require a genotype–phenotype linkage. Recently, a nanopore sequencing technique has allowed for the direct sequencing of a 26-amino-acid peptide sequence [63]. This novel technique uses azide click chemistry to link a DNA strand to the C-terminus of the protein, where the DNA is then pulled through a bacterial pore using a DNA helicase. As the nanopore system detects charge variation across the pore to detect molecular signatures, it may be possible that it can also sequence protein phosphorylation or glycosylation. The technology has been used to distinguish unphosphorylated, monophosphorylated and diphosphorylated protein states [64]. Technologies such as this could be the next generation of template-free library screening for peptides.

Approaches to improve the efficiency of labelled molecular library screening are becoming increasingly inventive. Microfluidics platforms are one of the most prevalent tactics to improve speed and efficiency of biopanning. Generally, these platforms are designed to reduce sample volume, create controllable fluid dynamics (uniform shear stress and velocity gradient), improve panning efficiency, and allow for automation of the biopanning process. It is important to be able to control fluid dynamics in panning systems, as mechanical force variation from shear stress induced in fluid flow affects peptide and antibody affinity for targets significantly [65,66,67,68]. Therefore, microfluidic channel fluid dynamics for library panning is an extensively studied field. Various modes of microfluidic panning exist from surface-adhered protein targets to microfluidic panning and screening using whole cells. Biopanning of phage-displayed proteins against surface-adhered cell cultures in microfluidics platforms has been shown to have a 600-fold greater level of enrichment when compared to conventional cell suspension-based biopanning [69]. Single-cell screening approaches such as Celli*GO* [70] use droplet microfluidics to perform high-throughput screening of immunised mouse-secreted IgG activity against soluble or membrane-bound antigens on a cell-by-cell basis. Other novel approaches include the use of alternating electrical current electrohydrodynamic (AC-EHD) flow, applied to disproportionally spaced gold electrodes, to improve microfluidic mixing at the target-phage library interface [71]. This phage library screening technique termed “PhageXpress” has been reported to require only a single biopanning step with AC-EHD flow, in combination with next-generation nanopore sequencing to achieve sufficient physical enrichment for isolation of target-binding phage clones. AC-EHD flow, used also in on-chip diagnostics [72,73,74,75,76,77,78], is proposed to reduce the “non-slip” property of fluid as it approaches a surface—a phenomenon described by Poiseuille’s law [79]—which can perturb microfluidic technologies. On-chip application of electrical potentials has also been used to produce digital microfluidics, where for example, electrowetting on dielectrics has been used to automate all steps—including counter selection—in multiple rounds of phage biopanning to produce a competent target-binding peptide [80].

### 2.3. In Silico Techniques

Bottom-up in silico (computational) approaches have grown from complementing and augmenting in vivo and in vitro experimental techniques for molecular discovery. Techniques either search the theoretical chemical space to produce a list of candidate molecules that can be verified for function with wet-lab experiments or search already wet-lab-characterised molecules to find suitable structural templates for novel molecular design. More recently, computational methods have been used to begin to design molecules de novo, using data available online compiled from years of wet-lab experimental work. This is a movement towards more top-down molecular discovery approaches and will be described in more detail in the latter part of this review. In this section, some of the prominent bottom-up molecular discovery techniques will be briefly discussed.

It has become common for chemical libraries to undergo screening for activity against therapeutic targets using in silico approaches, as it is infeasible for in vitro synthesised libraries to represent the full extent of the possible drug-like molecules [81]. The accessible chemical space has been estimated to include at least 10^60^ [82] observable molecules, which is currently an impractical number of molecules to attempt to screen with wet-lab approaches [83]. In silico approaches enable a wider chemical space to be explored, where databases of online virtual libraries can computationally model chemical-target docking, enabling the screening of vast libraries of potential chemotherapeutics prior to synthesis for experimental validation. Recently, it has been reported that virtual libraries of up to 11 billion compounds can be screened [84]. Generally, the target structure is converted into a representation that can be used by the docking software. The binding pocket structure is then optimised using control calculations with data from known interactions. After screening and docking the virtual compound library, the top hits are selected for experimental validation [81]. Virtual screening enables potential therapeutic molecules to be narrowed down to a smaller, more experimentally viable number of compounds to be synthesised and screened. For example, the ZINC free online database [85] currently contains over 230 million compounds that can be readily synthesised, which are also available on the database in 3D enumerated formats that can be virtually screened and docked against structure-solved targets. Quantitative structure–activity relationships (QSAR) or similar methods are frequently used for optimising lead compounds after screening is performed, as they help elucidate the relationship between molecular structure and activity and can help to enhance this. These models seek to find a mathematical relationship between structure of chemicals and their various properties using regression and classification techniques [86,87]. Molecular docking programs also allow for predictive capability for interactions of known drugs with other targets, which can be a useful drug repurposing tool or off-target toxicity prediction tool.

Most current in silico techniques for discovery of biologically generated molecules (peptides and antibodies) are designed to enhance and compliment wet-lab experimental data by building on and enhancing known structures. For example, computational modelling using data from BLAST searches of custom online databases and PDB has been used to assist prediction of the in vitro developability of antibodies [88] or to model antibody variable fragments (Fv) to predict protein–protein and protein–antibody interactions [89,90]. Deep generative computational models—for example, long short-term memory networks (LSTM)—have been used to produce improved binders from a phage-displayed single-chain variable antibody fragment (scfv) panning campaign. Saka et al. (2021) [91] used an LSTM network to produce antibodies that were reported to have over 100-fold improved affinity compared to binders that were affinity matured purely with experimental work. Input data were generated from next generation sequencing of enriched phage sequences obtained from multiple biopanning rounds against Kynurenine protein. Codons were translated into amino acids and then encoded into a format suitable for data processing. The LSTM model was then trained from this enriched sequence data. New sequence generation of proposed improved binders was then performed, where for each new sequence, the negative logarithm of likelihood (assumed to correlate with binding affinity) was calculated to select new high-affinity binders. Other studies include the use of statistical analysis [92] or machine learning (ML) approaches [93] to enhance enzyme activity, where enzymes containing genetic mutations or variations were screened based on sequence for altered enzymatic activity. The resulting sequencing and activity data enabled the synthesis of new enzymes that had greater activity than to the original enzyme.

Whilst using the available molecular docking programs enables a far higher-throughput approach to screening the vast chemical space, these approaches are not yet able to comprehensively flip therapeutic discovery to a top-down approach. Although chemical compound structures are more easily computationally predicted than proteins, molecular docking still requires the input of a high-resolution image of the target structure (preferably ligand-bound) [81]. Protein Data Bank (PDB) [94], one of the main online databases for protein sequence–structure information, currently contains 184,700 released protein structures as of 7 December 2021. However, only 2666 of these released structures are from *Homo sapiens*, where at present, there are about 79,000 known proteins in the proteome as described in the online database UniProt [95]. Furthermore, only 71,489 of these released structures are within a 2 Å resolution. With the currently minimal knowledge of protein structures in the human proteome, in silico molecular docking and screening cannot yet be performed on the entire human proteome. As discussed previously, current gold-standard technologies for solving protein structure are laborious and time consuming, and it can be particularly difficult to obtain accurate representations for some targets [96]. Therefore, the advent of new technologies which are beginning to accurately predict protein structure from a sequence is a particularly exciting prospect for therapeutics development.

## 3. The Limitations of Bottom-Up Approaches

The current pipeline for taking new therapeutic molecules from discovery to clinic involves an incredibly lengthy, complex, and expensive process. This is in part because drug development necessitates collaborations between many fields of expertise, from chemists and biologists to process scientists, clinicians, intellectual property and industry specialists and partners. It has been estimated that the median capitalized development and research cost to bring a new medicine to FDA approval in the United States between 2009 and 2018 was about USD 985 million, with projected costs expected to be in the billions [97]. The median clinical development time for FDA-approved drugs has remained stable at approximately 8.3 years for the past decade [98]. The significant time, expertise and cost associated with drug development usually translates into expensive out-of-pocket costs for patients [99,100]. As such, the process is not as simple as performing a molecular library screen and finding a new molecule that binds to a target.

There is an extraordinary amount of resources used in the current drug development pipeline. It takes years of work in the laboratory to first study and understand the biology of new therapeutic targets. Vast arrays of possible effector molecules are then screened against these new targets and narrowed down to a small cohort of binders, which are then characterised and further tested. Understanding the mechanisms and key targets for certain diseases can take entire careers to research and discover; thus, the next hurdle of developing a new therapeutic and taking it to clinic remains. For example, high-throughput molecular screening approaches for new drug discovery, such as phage-displayed library panning and enrichment [25,26] take molecular libraries, many orders of magnitude in size, and essentially “find what sticks” when searching for a new molecule to bind to or react with a target. In vivo techniques may also be used, where generally a target molecule is inoculated into an animal, and antibodies are produced naturally by the animals’ immune system. Advances in this field mostly involve using alternative molecular display vectors, improving the efficiency of molecular panning and enrichment, or genetically engineering animals for better antibody generation. However, these approaches are still generally inefficient. The next step is to characterise these molecules (structure, mechanism of action, solubility, and permeability), optimise the binding properties, and then validate the binder in in vitro and animal models. The gold-standard techniques for solving protein structure—X-ray crystallography [101] and NMR spectroscopy—are capable of solving the crystallised structure of proteins with incredible detail but remain complex and time consuming [96]. In some cases, particularly for membrane-bound proteins, the crystallisation step required to image the protein is not possible. Usually, during the characterisation process, the drug is also tested in an appropriate animal model. In 2015, there were 192.1 million animals estimated to have been used for laboratory purposes globally [102]. There have been questions over the efficacy of the use of animals in drug testing, where occasionally, drugs deemed safe and effective in animals have proven toxic and ineffective in humans [103]. A summary of some of the advantages and disadvantages to in vivo, in vitro and in silico approaches can be found in Figure 5. Thus, current techniques to drug discovery can be costly, and questions over their effectiveness highlight a need to search for alternative approaches.

Once a drug has been isolated, it must then make it to the clinic. If enough funding and support is received, and a candidate molecule has been successfully isolated, trials can then begin in a small group of patients. If the molecule passes safety and efficacy tests, then the trials can expand to larger groups of patients. Because it takes a significant amount of funding to develop new therapeutics, it becomes a substantial gamble and barrier for investors on whether a potential new molecule will have clinical value compared to existing treatments. Unfortunately, this means that many drugs may not make it to the clinic simply because they have not received the required financial support. Once the therapeutic has been shown to have efficacy, it must still be able to be manufactured and distributed in a form that it is conducive to its use (e.g., thermostability or lyophilisation). Therefore, it is of significant interest to develop new techniques for therapeutics development that give investors and prescribers greater confidence in the efficacy of the product. Greater confidence that the molecule is going to behave in a more predictable way through patient trials and in the market will help to streamline the development process, reducing barriers to investment and delivery to patients.

The time and costs are not the only negatives associated with the slow and expensive drug development pipeline. Current limitations in the discovery pipeline means that current disease treatment tends towards more generic, rather than personalised approaches, which can affect patient experience and outcomes. With existing technologies, it is not yet feasible to develop molecular therapeutics on a patient-to-patient basis, although individual patient screening and characterisation and the use of global therapeutics that are tailored to a patients’ clinical presentation is generally regarded by clinicians as crucial [104,105,106]. This has implications for patient outcomes, where significant inter- and intra-patient variability [107] means that available therapeutics are not only expensive [100] but can be “unreliable” in that they can cause unforeseen adverse effects and varied or limited impacts on a disease [108,109,110,111,112]. With the example of cancer, global treatments include classic chemotherapy and targeted treatments such as monoclonal antibodies (mAbs) or peptides. Classic chemotherapy involves the use of cytotoxic chemicals, including folate and purine analogues, inhibitors and promoters of microtubule polymerisation, and DNA damaging agents [113]. These cytotoxic agents act by interfering with cellular division, which affects not only rapidly dividing malignant cells but also any other cells in the body that are undergoing division, such as haematopoietic and immune cells [113,114,115]. Freely circulating chemotherapy drugs often require administration in high doses, as they accumulate in high concentrations in the kidney and have short circulation times [116]. This results in a need for higher drug doses, which is problematic because classic chemotherapy has limited specificity and even low doses can cause systemic toxicity [117,118], whilst often resulting in only minor tumour response and sometimes tumour drug resistance [111,119,120].

Targeted therapeutics such as mAbs have high specificity for target antigens and have been employed as therapeutic agents for the past two decades [113]. Antibodies can impart therapeutic benefit by more specifically binding to targeted cells presenting specific antigens and marking them for antibody-dependent cell-mediated cytotoxicity (ADCC), complement-dependent cytotoxicity (CDC), or antibody-dependent cellular phagocytosis (ADCP) through opsonisation of target antigens [121]. Antibodies can also bind to targets and inhibit downstream functions through neutralization [113], and they have reportedly been found to induce cell differentiation and induce immune responses to target cells [122]. More recently, conjugation of antibodies and targeting peptides via linker chemicals to cytotoxic or radioactive payloads has allowed for precise delivery of therapeutic cargo to specific target cells [123,124]. Radiolabelling allows for either imaging or treatment (dependent on dosage), where uptake can be visualised through positron emission tomography (PET) scans. This mode of treatment has been coined as “theranostics” (therapeutic diagnostics). Whilst these targeted therapies are more precise and are less likely to impact off-target cells, significant inter- and intra-patient biological variability [107] means that treatments can cause unforeseen adverse effects and varied or limited impacts on a disease, significantly affecting patient outcomes [108,109,110,111,112]. Creating a pipeline where the off-target effects of therapeutics could be more accurately predicted and where therapeutics are more precisely designed for each targeted effect would be invaluable to patient outcomes.

Top-down approaches, where we can design a therapeutic from scratch and be able to predict molecular interactions, would have a massive benefit to streamlining the development pipeline and ultimately to patients. If we could design a therapeutic from scratch to specifically interact with a desired target, then it is possible that we could design novel therapeutics or modify existing ones with their specific interactome in mind. This would enable better prediction, and possibly elimination, of off-target toxicities in patients and better tools for clinicians to predict which patients to prescribe and which patients not to prescribe. For the same reasons, a top-down approach may help to streamline many areas of the therapeutics development pipeline, not only by improving molecular design but by making products more likely to be competitive in the market. Increased confidence in and better prediction of drug performance shifts the risk of drug development to earlier in the pipeline, where there is minimal collateral damage incurred upon the failure of a new product. This could help reduce barriers to financial investment and regulatory approval at later stages of the pipeline by increasing the likelihood and predictability of drug success.

## 4. Towards a Top-Down Approach

### Machine Learning for Predicting Molecular Structure and Complex Formation

Recent advances in computational science are beginning to make the paradigm shift to a top-down approach for molecular discovery possible. Major developments in high-throughput technologies, including next generation sequencing, protein expression and protein characterisation, protein structure deconvolution and computational advances, are the foundation for shifting to top-down technologies. The wealth of information available pertaining to molecular sequence and structural information in online databases has accelerated computational technologies to the forefront of top-down molecular discovery approaches. Whilst many computational approaches have largely focussed on improving known target-ligand interactions or are designed to complement wet-lab techniques, recently, there has been a shift towards developing entirely de novo molecular designs.

A range of de novo approaches to molecular design has been developed to date. Deep and reinforcement learning has been used to generate novel chemical libraries that have desired properties. A novel computational strategy call ReLeaSE (Reinforcement Learning for Structural Evolution) integrates a generative and a predictive deep neural network to generate libraries of de novo structures. These libraries can be geared towards having certain levels of structural complexity, or a specific range of physical properties (melting point, hydrophobicity), or to have inhibitory activity against certain targets. Specifically designing libraries in this manner allows molecular libraries to be targeted towards certain functions and increases the likelihood of finding a molecule with desired properties in wet-lab experiments. The process of obtaining a new binder using this method is not strictly top-down, as the new libraries still need to be specifically screened against the target of interest. However, they still use de novo molecular generation techniques to create desired properties; thus, this could be viewed as a top-down approach in this aspect.

ML algorithms trained using sets of known structures of proteins available online on PDB and other databases have recently made significant advances in predicting protein folding [125,126,127]. DeepMind’s AlphaFold 2 has received a considerable amount of media attention for having “solved” the 50-year-old grand challenge of using the nucleic acid sequence of a protein to predict protein folding. In the 14th Critical Assessment of Protein Structure Prediction (CASP14) in 2020 [128], AlphaFold 2 achieved computational prediction of protein folding to near-experimental accuracy (all-atom protein structure prediction from within 1.2 to 1.6 Å at a 95% confidence interval, where the width of a carbon atom is approximately 1.4 Å) from the analysed sequences [126]. The model’s training uses a set of protein structural and sequence data, and multiple sequence alignments (MSAs) of the original sequence against proteins from many species, giving evolutionary information on sequence conservation. These data were derived from the Protein Data Bank [94] and Uniclust30 [129], and MSAs were generated using publicly available software. The building block of the novel neural network—“Evoformer”—produces processed representations of the MSA of the presentation of residues across different species and a processed pairwise representation of the structural relation between residues (*N*_res_ × *N*_res_ array) that are refined as they progress through the model. The final paired and MSA representations are then fed into a model that gives 3D structure in terms of rotation and translation for each residue of the protein, and further iterative refinement processes in the model produces an output of predicted protein structure.

There is a range of groups that have produced variations or alternatives to the technology, particularly since the second iteration of AlphaFold from the CASP14 competition. The team behind the second-best protein structure prediction method after AlphaFold, RoseTTAFold [130], which can operate within a fraction of the time taken by AlphaFold (~10 min on a single GPU as opposed to days with several GPUs), has since produced a combination of AlphaFold and RoseTTAFold to predict protein complexes [131]. Using a paired MSA, they identified 1505 likely-to-interact proteins to form complexes in yeast *Saccharomyces cerevisiae*, 699 of which had been structurally characterised and 700 of which were supported in the literature. However, these are predictions of stable protein complex formations, which would likely have highly conserved and stable pMSAs through different species, due to the nature of the interaction being under significant selective pressure. More transient, weaker interactions that are under lower selective pressure were noted by the authors as likely to be difficult to predict, as well as binary protein complexes from high-order obligate protein associations. It was also noted that predictions would be inaccurate for species-unique protein complexes and that there would be an overrepresentation of hydrophobic or amphipathic domains. As well as this, recent approaches involving deep network hallucination have investigated the generation of new proteins with entirely novel backbone structures and amino acid sequences [132].

Other groups have made significant steps towards de novo design of target-binding proteins given nothing but the target structure. The design of proteins given no other information (such as homologous target-binding proteins) other than the target structure has been a major challenge for in silico therapeutics development. As previously described, it is challenging to predict interactions that are more transient, non-covalent and under lower selective pressure. The David Baker lab at EMBL-EBI is one of the first groups claiming to be able to perform de novo design for more transient interactions of protein binders and targets. In this work, the group has moved beyond using a native or homologous protein scaffold and previously identified hot-spot residues to identify and predict proteins with favourable and more transient interactions [133]. This approach begins by performing target docking of a library of “disembodied” amino acids and creating what is termed a “rotamer interaction field” (RIF). This RIF is the stored backbone coordinates and binding energies of billions of amino acids that are screened in this docking approach that have favourable more transient (hydrogen bonding and non-polar) interactions. This RIF is used as a field which targets docking of large protein scaffold libraries that can be more rapidly scored (due to reduced algorithmic complexity created through the RIF) based on only backbone amino acid coordinates. Focussed screening and motif clustering and selection allows the deconvolution of motifs that are predicted to bind to the target through hydrogen bonding or non-covalent interactions. Whilst the RIF is designed to attempt to reduce the required computational power, the process still requires many computationally demanding steps, and there is a low success rate for producing designs that bind the target successfully. Even among molecules that were found to bind in some capacity to targets, it was still required to make amino acid substitutions to create a strong binder. Therefore, whilst this approach is one of the first promising solutions to de novo design of therapeutics, more work is required to make this method deliverable.

Spirited claims have been made that the recent advances in machine learning models are one of the greatest developments in structural biology in the past 20 years, which is also one of the biggest achievements in artificial intelligence in recent history [134]. Some have even asserted that that AlphaFold’s predictions have virtually solved the structure of the entire human proteome [135]. Since CASP14, the AlphaFold team in conjunction with EMBL-EBI has recently released their predictions for 98.5% of the human proteome through building on the AlphaFold model, where they state to have predicted 58% of proteins with confidence and 38% with high confidence [136]. As previously described, ligand docking is already used to computationally model how chemical therapeutics interact with structurally solved targets and to screen libraries of potential chemical therapeutics against these targets. Therefore, the implications for being able to predict protein folding without the need for experimental validation of protein structures—which is one of the biggest current bottlenecks to in silico approaches to therapeutics discovery—would quickly and significantly extend the possibilities in the field of therapeutics discovery. It is possible that eventually all library screening could be performed in silico without the need for experimental determination of target—or protein and antibody library—structures. In addition, instead of a bottom-up screening approach where an investigator is trying to “find the needle in the haystack”, you might be able to flip this to a top-down approach, where one could build a molecule with the exact characteristics you desire entirely de novo using computational modelling.

Whilst the prospects of this computational achievement are exciting, it remains a question of how realistically useful and reliable these models are in their current state, particularly for molecular discovery applications. Foremost, this technology is not yet user friendly or easy to access by anyone who is not an expert in the field. For example, whilst the open-source code for AlphaFold has been released and is publicly available (https://github.com/deepmind/alphafold, accessed on 28 February 2022), the value in ML lies not in the code but in the training dataset. ML algorithms will have markedly different outputs depending on the data they are trained on. Whilst it is possible to recreate the AlphaFold training dataset, it remains highly computationally challenging and time consuming to build and mimic. This means that whilst the code may have been released for these ML algorithms, it is not in a form that allows this technology to be readily accessed by researchers to predict new structures or interactions. As well as this, whilst data for protein structure are readily available, the complexity of molecular interactions and a limit to online datasets specific to molecular interactions and therapeutics make AI for molecular discovery and interaction predictions challenging. Regardless of the challenge of compiling a robust and accessible drug-related database, proteins are constantly in motion in cells [137]. Not only are they being altered by post-translational modifications, but they are changing conformations depending on the function being performed and ligand-bound states. Generally, to model ligand docking, the ligand-bound (holo) protein structure is preferred to the unbound (apo) structure, and there is some question as to whether current AI protein structure predictions can differentiate between holo and apo states. Some studies have shown that peptide-bound protein complexes could be predicted accurately by concatenating the sequences of receptors and peptides using a poly-glycine linker [138]. Google’s Deepmind has also recently released the source code for their “AlphaFold-Multimer”, which has been reported to be able to accurately predict heteromeric protein docking interfaces in 67% of cases, and homomeric interfaces in 69% of cases, but were not able to predict antibody interactions [139]. However, other researchers have remained uncertain about the accuracy of predictions in protein active sites, as when AlphaFold and RoseTTAFold predictions were overlayed onto the experimentally solved structures of 20 GPCRs, only about 50% of the predicted structures were useful [140]. AlphaFold is expected to have poor predictive capability for the intrinsically disordered active sites or those that are unstructured in isolation [140]. It is also not expected to be able to capture the protein destabilising effects of point mutations, which would be important for being able to accurately model disease states.

Although there are many caveats to in silico predictive technologies that need development, there is still much promise in the field. Similar issues to accurate structure and protein interaction prediction exist with any protein structure imaging technique. Whether an image is obtained in the holo or apo state, it is still only a snapshot of the possible conformations that this protein could undertake. As previously mentioned, there are also a significant number of protein structures in PDB that are relatively low resolution due to limitations to wet-lab experimental approaches; thus, ML techniques such as AlphaFold could help to improve the ability to perform molecular docking with these structures. Post-translational modifications, such as glycosylation, and flexible, less constrained domains are also still difficult to structurally determine using X-ray crystallography [141]. It may also be possible that ML models could be trained without the need to obtain protein structures. Early studies have suggested that it is possible to train deep generative models exclusively on antibody sequence data to design new epitope-specific antibodies [142]. As previously discussed, sequence data can be used to make statistical inference or train ML algorithms, enabling the improvement of enzymatic activity or antibody affinity [91,92,93]. If sequence data could be used in lieu of structural data to train ML algorithms to predict new therapeutics, they could significantly expand this field of therapeutics discovery and off-target prediction. Sequence-based data are significantly more accessible than structural data because experimental validation of the interaction can be derived from sequencing and relatively simple laboratory tests such as ELISA rather than X-ray crystallography or cryo-EM.

It is not yet clear how this new AI technology can be adopted by drug developers, and in which role it will play in therapeutics development. It can be hypothesised that the technology will streamline and accelerate the journey to progress a therapeutic from discovery to clinic. If successful, the technology could reduce the time taken to obtain new therapeutics, by reducing or perhaps eliminating the need for bottom-up or library screening approaches. Blue-sky thinking would encourage the idea that we may begin to create drugs on an individual patient basis. Potentially, the technology could allow for identification and screening of off-target binders that produce toxicity for patients by allowing whole proteome screening. Being able to reduce the risk of patient toxicity would certainly improve investor confidence that the drug candidate will be successful in clinic, as well as decreasing the likelihood of failure through clinical trials. These things would allow for the acceleration of therapeutics through the development journey. Whilst the recent predictive technologies are a shift in the top-down approach to molecular discovery, in the current state, they are not yet positioned to replace the existing gold standard in in vivo and in vitro techniques. As well as this, biological diversity is thus large that the amount of information that would need to be processed for AI to predict off-targets for a single existing drug for each patient with high accuracy would be vast, let alone creating new drugs on a patient-by-patient basis. Predictive models such as AlphaFold still require significant development and further prospective validation experiments, but existing data are promising.

Before computational approaches can enable a top-down approach to drug discovery, a united effort between all areas of bottom-up therapeutics discovery will likely be required. Expansive, high-quality wet-lab data are generally considered as critical for the development of successful ML algorithms for molecular discovery [143]. Large volumes of binding motif data are highly valuable to predict paratope–epitope features [144,145]. Therefore, the use and further development of bottom-up wet lab techniques designed to feed into more extensive online databases and computational analysis would be likely to help drive predictive outputs for top-down approaches. If it is possible to predict paratope–epitope interactions from sequence only, then it may be suggested that high-throughput wet-lab experimental techniques that can quickly and consistently identify binder-target sequences—not only holo structures—of molecular interactions in a high-throughput manner would be especially useful. This means that high-quality experimental design and consistent and high-throughput assay technology that can be used to create data from a broad range of samples and conditions will be a driving factor behind enabling top-down therapeutics design. Selected techniques will need to minimise noise and batch effects (e.g., variation in data from performing the same experiment in two different laboratories), as this would likely have a significant effect on ML-based predictions.

## 5. Conclusions

The recent advances in therapeutics discovery have the potential to be highly beneficial to all areas of the drug discovery journey. In summary, the critical points from this review article are:There is a clear need for a more efficient pipeline for the journey of therapeutics, from discovery to the clinic, as highlighted by the recent novel coronavirus pandemic. Emerging AI technologies may enable a smoother transition through the various stages of the pipeline, including reducing barriers such as regulatory hurdles and market performance.Bottom-up techniques are slower and more resource exhaustive than top-down techniques and, due to the nature of the approach, requires much downstream characterisation and validation. It is recommended that we should continue to improve the quality of data generated from bottom-up technologies. This will be a critical step to move to top-down technologies, as these data will feed these new approaches.The earlier that the risk of failure of the development of a new drug is addressed and predicted, the more likely it is that the drug will successfully and quickly reach the clinic with minimal losses. It is recommended that we move towards top-down approaches to drug discovery that enable stronger understanding of molecular behaviour, as investors are more likely to support a drug that they are confident will be successful and competitive in the market.It is hypothesised that access to more personalised medicine will enable clinicians to effectively compete with the biological intricacies of complex disease.Emerging top-down AI technologies are improving the prediction of molecular structures, molecular behaviour and optimal de novo drug design. It is recommended that we continue to incorporate and improve these techniques into therapeutics discovery, as they will help to streamline the development pipeline, although further work is needed to contend with the massive complexity of biological systems.Encouraging the sharing and improved accessibility to ML approaches for people who are not specialists in the field may help to drive innovation and discovery with new ideas and perspectives.

## Figures and Tables

**Figure 1 life-12-00363-f001:**
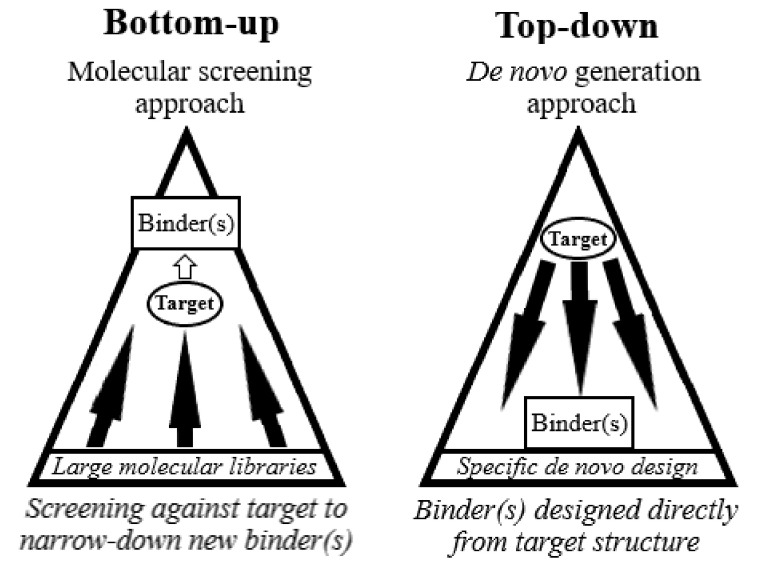
Bottom-up and top-down approaches to ex vivo molecular discovery. Bottom-up approaches generally involve screening broad libraries of candidate molecules against a target of interest and narrowing this down to a single best binder. The top-down approach is to design and generate new binders directly based on the structure of a target molecule.

**Figure 2 life-12-00363-f002:**
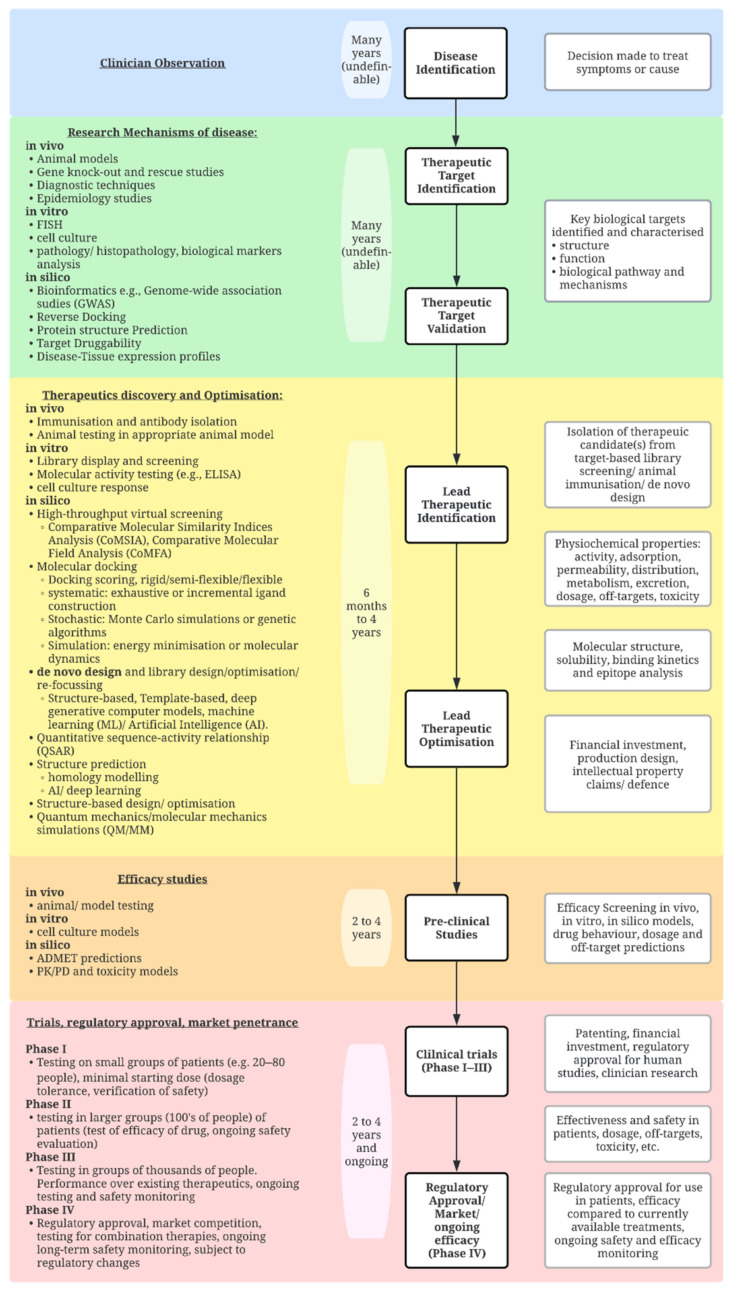
The current journey of therapeutics from discovery to the clinic (top to bottom), with prominent examples given of in vivo, in vitro and in silico approaches. A broad approximation of time taken for each stage of development is provided. (Created with lucidchart.com).

**Figure 3 life-12-00363-f003:**
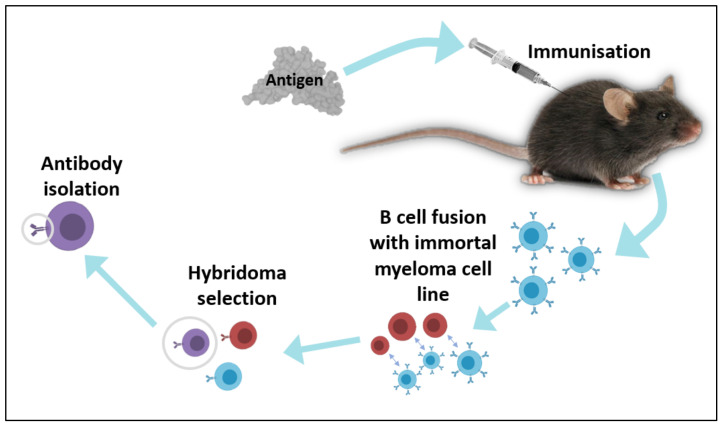
Traditional process of in vivo antibody isolation. First, animals are immunised with an antigen of interest. After an immune response has been mounted, B cells are isolated from the spleen of the animal and fused with an immortal myeloma cell line. Hybridomas are selected and screened for their activity against the antigen of interest. Target-binding clones are retained and used to obtain purified antibodies.

**Figure 4 life-12-00363-f004:**
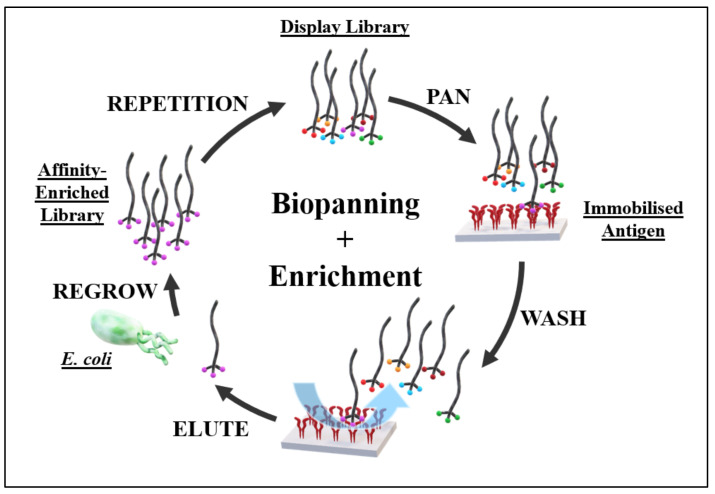
Traditional library biopanning and enrichment against a target. Broadly, the genetically encoded library is incubated against a surface-bound antigen—the biological target of choice. Unbound phage are washed away whilst target-bound phage are reserved. Target-bound phage are then eluted from the immobilised antigen and regrown in *E. coli*. The now affinity-enriched phage cohort is then fed into the next cycle of biopanning.

**Figure 5 life-12-00363-f005:**
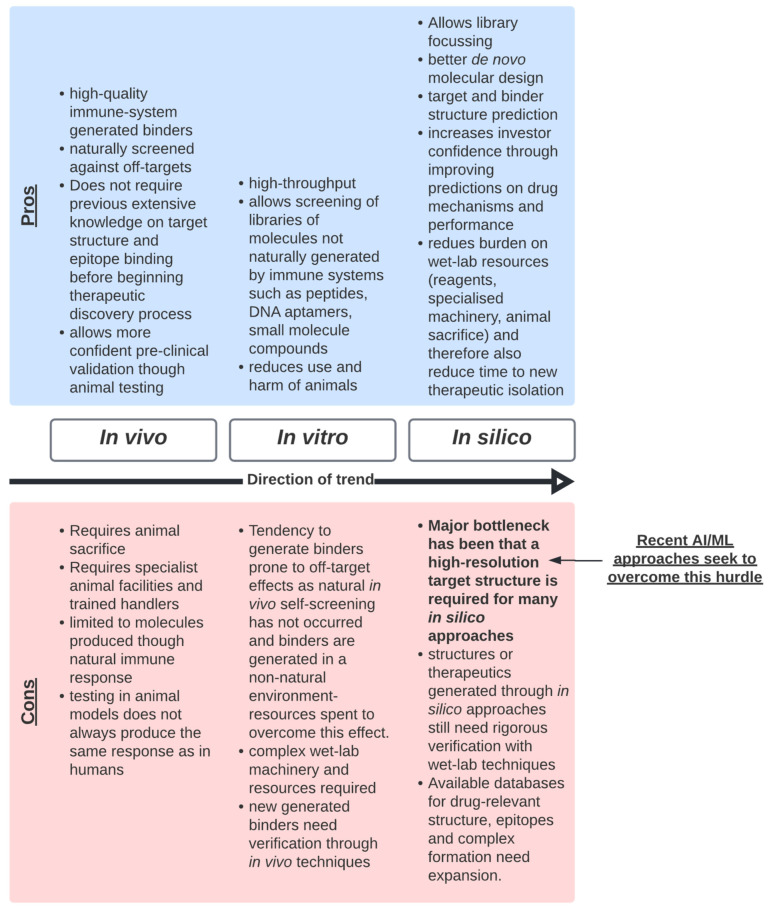
Some of the major advantages and disadvantages to each broad approach to therapeutics discovery, with the trend moving towards increased use of in silico technologies.

**Table 1 life-12-00363-t001:** List of major approved SARS-CoV-2 treatments by select regions as of 4th February 2022.

Drug Name	Date of TGA Provisional Approval in Australia (TGA)	Date of Emergency Use Authorisation in United States (FDA)	Date of Authorisation in European Union (EMA)	Date of Approval in Japan (PMDA)	Date of Authorisation in Canada (Health Canada)
Remdesivir	10 July 2020	5 February 2020 (eligible patients)	(conditional) 3 July 2020	7 May 2020	27 July 2020
Sotrovimab	20 August 2021	26 May 2021	17 December 2021	27 September 2021	30 July 2021
Casirivimab and Imdevimab	15 October 2021	21 November 2020 (removed until further notice from 24 January 2022 as of 4 February 2022)	12 November 2021	19 July 2021	9 June 2021
Tocilizumab	1 December 2021	24 June 2021	7 December 2021	-	-
Regdanvimab	6 December 2021	-	12 November 2021	-	-
Molnupiravir	18 January 2022	23 December 2021	-	-	-
Nirmatrelvir and ritonavir	18 January 2022	22 December 2021	28 January 2022	-	17 January 2022
Baricitinib	-	19 November 2020	-	-	-
Bamlanivimab and etesevimab	-	25 February 2021 (removed until further notice from 24 January 2022 as of 4 February 2022)	-	-	20 November 2020 (Bamlanivimab only)
Tixagevimab and cilgavimab	-	8 December 2021	-	-	-

**Table 2 life-12-00363-t002:** Prominent types of labelled library display.

Display Modality	Library Molecule Types	Maximum Library Size (Unique Clones)
DNA-displayed chemical library [36]	Single pharmacore (DNA-recorded synthesis), Dual-pharmacore.	10^11^ [49]
Phage Display (pIII coat protein most common fusion) [25,26]	Peptides, ScFv, Fab, sdAb/nanobodies.	10^11^ [50]
Yeast Display (Aga1p + Aga2p most common) [51]	Peptides, ScFv, Fab, sdAb/nanobodies, whole antibodies	10^9^ [52]
Bacterial Display (Ipp + ompA, PAL, AhaA and intimin β-domains, APEx-NlpA or g3p, MAD-TRAP) [53,54]	Peptides, ScFv, Fab, sdAb/nanobodies, whole antibodies	10^10^ [54]
Mammalian cell display (fusion to transmembrane domain such as H-2Kk or PDGF receptor) and secretion (LoxP site inclusion on membrane anchor domain) [55,56]	Peptides, ScFv, Fab, sdAb/nanobodies, whole antibodies	10^9^ [57]
mRNA display/cDNA display [34]	Peptides, ScFv, Fab, sdAb/nanobodies	10^15^ [58]
Ribosome Display [33]	Peptides, ScFv, Fab, sdAb/nanobodies	10^15^ [58]

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
