# Peer review of "Next-Generation Molecular Discovery: From Bottom-Up In Vivo and In Vitro Approaches to In Silico Top-Down Approaches for Therapeutics Neogenesis"

_life, 2022, doi:10.3390/life12030363_

Round 1

Reviewer 1 Report

I appreciate the authors as they addressed all of my comments. I have no further comments. I'd like to recommend the paper for possible publication in LIFE.

Reviewer 2 Report

The manuscript entitled
"Next-Generation Molecular Discovery; Progressing from bottom-up
immunisation and library screening approaches to in silico top-down
approaches for therapeutics neogenesis" has incorporated the suggested inputs in the revised version. In my opinion the manuscript is good to go for the publication.

This manuscript is a resubmission of an earlier submission. The following is a list of the peer review reports and author responses from that submission.

Round 1

Reviewer 1 Report

This review article describes current approaches to molecular discovery, their limitations, and possible future directions. Authors categorized molecular discovery processes into “bottom-up” and “top-down” approaches. They defined “Bottom-up” approaches as the methods that use in vivo animal antibody generation or in vitro/in silico library screening while “top-down” approaches as the methods that de novo design therapeutic molecules from scratch. In the paper, the authors describe the current “bottom-up” approaches by grouping them into in vivo, in vitro, and in silico approaches. Those “bottom-up” approaches tend towards more generic, rather than personalized approaches because it takes a significant amount of time, labor, and funding which makes it not feasible to develop therapeutics on a patient-to-patient basis. With recent progress in protein tertiary and complex structure prediction based on AI, authors claimed that it might be possible to make an in silico top-down approach in the not-too-distant future although it requires further development. This review paper summarizes the current status and provides meaningful perspectives in the field, but the manuscript itself should be improved and re-organized to make the message much clear.

Major comments:

  1. I’d suggest adding Figure 1 which illustrates what are the bottom-up and top-down approaches so that readers can get their definition clearly by just looking at the first figure. It would be even better if the reader can understand how the trend is changing (from bottom-up to top-down) through figure 1.
  2. Figure 2 (LSTM-based antibody generation) seems unnecessary. If readers are interested in that method, they probably check the referenced paper. Instead of Figure 2, I’d suggest adding a table or figure that summarizes current in silico approaches.
  3. Figure 3 summarizes the therapeutic discovery process well, but it would be even more informative if the authors add how long each step takes to make it more supportive to the main text (claiming the current bottom-up approaches are time/labor consuming).
  4. Figure 4 is unnecessary. I think there’s no need to explain detailed AlphaFold architecture in this review.
  5. It’s a little bit confusing which direction authors want to emphasize. Is it from experiments to in silico approaches (both bottom-up/top-down) or from bottom-up to top-down approaches? With recent advances in AI-based protein structure prediction, in silico bottom-up approaches could also be boosted as the authors pointed out in the manuscript.
  6. There are several top-down approaches developed so far [e.g., fragment-based drug discovery, de novo binder design (Longxing Cao et al, Science (2021)), deep reinforcement learning-based de novo drug design (Mariya Popova et al, Science advances (2018)), etc]. Those methods should be described in the manuscript.
  7. It would be great to have a figure or table that summarizes pros and cons of in vivo, in vitro, and in silico approaches described in section 2.
  8. In section 4, even though the subtitle is “Machine learning for predicting structure and protein complex formation”, there’s no mention of complex structure prediction. It seems better to move yeast interactome prediction and AlphaFold-multimer part to section 4.
  9. It explains AlphaFold2 architecture in detail (from line 437 to line 447), but it seems unnecessary for this review article, and there’s no further argument development based on AlphaFold architecture. Please remove them or make them short.
  10. AlphaFold model is trained on protein structures in the Protein Data Bank, and the input MSAs were generated based on publicly available softwares (HHBlits, JackHMMER) and sequence databases (UniProt, BFD, etc). It’s straightforward to generate the training set for protein structure prediction equivalent to AlphaFold’s one (although it takes some time). I think, for molecular discovery, the main problem is lack of “large” drug-related databases such as antigen-antibody complex structures, protein-drug affinity data, etc. Unlike general protein structures, those data are too small and noisy to train deep learning models.

Minor comments:

  1. In line 19, please remove “and protein folding” because it’s a synonym for protein structure prediction.
  2. In line 22, it would be better to define what is a top-down approach.
  3. At line 97, I think V(D)J recombination is more commonly used than V, D, J recombination.
  4. In line 123, there is a missing “are”. Please correct it to “they are not as high throughput as”
  5. Although I think Figure 2 is unnecessary and should be removed or replaced, in the figure legend (line 262), it refers the wrong reference. It should be “reproduced from Saka et al 2021 [75]”
  6. At line 270, it might be better to replace “a brute-force approach” with “a brute-force wet-lab approach” to avoid any confusion to in silico approach.
  7. At line 456, it should be “EMBL-EBI”, not “AMBL-EBI”.
  8. At line 464~465, please add a brief comment on the limitations of AlphaFold models although they’re described in the next section. Having a brief comment here would give a consistent message (recent advances in protein structure prediction opens up a lot of new possibilities but requires further development).

Reviewer 2 Report

The manuscript entitled “Next-Generation Molecular Discovery; Progressing from bottom-up immunisation and library screening approaches to in silico top-down approaches for therapeutics neogenesis” by Kenny et. al., briefed bottom-up approaches in drug discovery program and summarized current scenario in development of top-down approaches. Overall, the objective of the manuscript is to provide the ongoing and emerging perspective of the drug discovery program. The manuscript can be of interest for the young readers to understand the basics of drug discovery and how the overall process of drug discovery works. I have minor comments to improve the manuscript:

  1. In consideration for the interest of broad readership it will be nice if authors can extend their table 1 for the approval of remdesivir based on continents instead of just to one country keeping in attention for the availability of the data/resources.
  2. Author mentioned Molecular discovery categorized into three generations, please cite the reference for this classification. It will be good to see the illustration for in-vivo technique as well in similar fashion like the author represented the in-vitro and make them Fig1a and Fig.1B. In addition, for section 2.3 “In silico technique”, it would be nice to highlight the Quantitative structure–activity relationship (QSAR) models as well and mention it briefly.
  3. Section 3: The section nicely briefed the overall process in drug discovery but it would be nice to provide some information for the type of animal use in testing/trials which can be of interest for younger readers. Although, authors mentioned in the flow chart for the clinical trials I-III, it is suggested to demarcate it in the text that which line/paragraphs comes under which stage of clinical trial.
  4. Please provide a box possibly in the conclusion section with concrete bullet points to highlight the merits and demerits of both the approaches to understand them in a glance.

Reviewer 3 Report

The manuscripts looks good and presented well. In terms of the results alone, this paper is interesting, and is clearly appropriate for inclusion in drug discovery. I recommend the paper for possible publication in LIFE.